# Impact of crop type on the greenhouse gas (GHG) emissions of a rewetted cultivated peatland

**Kristiina Lång, Henri Honkanen, Jaakko Heikkinen, Sanna Saarnio, Tuula Larmola, and Hanna Kekkonen**

Natural Resources Institute Finland, Latokartanonkaari 9, 00790 Helsinki, Finland

**Correspondence:** Kristiina Lång (kristiina.lang@luke.fi)

**Abstract.** Raising the water table is an effective way to abate greenhouse gas emissions from cultivated peat soils. We experimented a gradual water table rise at a highly degraded agricultural peat soil site with plots of willow, forage and mixed vegetation (set-aside) in southern Finland. We measured the emissions of carbon dioxide ($CO_2$), methane ($CH_4$) and nitrous oxide ($N_2O$) for 4 years. The mean annual groundwater table depth was about TS1 80, 40, 40 and 30 cm in 2019–2022, respectively. The results indicated that a 10 cm rise in the water table depth was able to slow down annual $CO_2$ emissions from soil respiration by $0.87\,\mathrm{Mg\,CO_2}$-C ha$^{-1}$. $CH_4$ fluxes changed from uptake to emissions with a rise in the water table depth, and the maximum mean annual emission rate was $11\,\mathrm{kg\,CH_4}$-C ha$^{-1}$. Nitrous oxide emissions ranged from 2 to $33\,\mathrm{kg\,N_2O}$-N ha$^{-1}$ yr$^{-1}$; they were high in bare soil at the beginning of the experiment but decreased towards the end of the experiment. Short rotation cropping of willow reached net sequestration of carbon before harvest, but all treatments and years showed a net loss of carbon based on the net ecosystem carbon balance. Overall, the short rotation coppice of willow had the most favourable carbon and greenhouse gas balance over the years ($10\,\mathrm{Mg\,CO_2}$ eq. on average over 4 years). The total greenhouse gas balance of the forage and set-aside treatments did not go under $27\,\mathrm{Mg\,CO_2}$ eq. ha$^{-1}$ yr$^{-1}$, highlighting the challenge in curbing peat decomposition in highly degraded cultivated peatlands.

## 1 Introduction

Cultivated peatlands are a major source of greenhouse gas (GHG) emissions globally (Strack et al., 2022). Conventional cultivation requires lowering the water table depth (WTD) which makes all peat above the drainage depth prone to microbial decomposition. Intensive management together with the high carbon and nitrogen content of peat makes agricultural peat soils the highest $CO_2$ and $N_2O$ emitters per unit area compared to any other land use types on peat soils (Maljanen et al., 2010). Their GHG emissions currently significantly diminish the net carbon sink of peat-rich countries, which can also be turned into an advantage: the climate change mitigation potential of drained peatlands is high (Humpenöder et al., 2020; Leifeld and Menichetti, 2018) and cost per mitigated unit of $CO_2$ equivalent low (Lehtonen et al., 2022).

WTD is the major controller of GHG fluxes from peat soils (Evans et al., 2021; Wilson et al., 2016). A global meta-analysis on water table manipulation studies showed that WTD explained most of the variation in GHG emissions, but, for example, the climate zone had some influence as well (Huang et al., 2021). Rewetting has been found to diminish the release of $CO_2$ and $N_2O$ from decomposition, but the switch from aerobic to anaerobic decomposition may change the ecosystem from a sink to a source of $CH_4$. However, the average increase in $CH_4$ emissions does not usually compromise the net GHG mitigation potential (Bianchi et al., 2021; Guenther et al., 2020; Mander et al., 2023), but data are needed to understand the factors regulating $CH_4$ emissions that can sometimes be high after rewetting (Nielsen et al., 2023).

Paludiculture, i.e. crop production in wet conditions on peat soils, is a GHG mitigation method that allows for slow-

ing down peat decomposition while still maintaining agricultural income from peatlands for the landowner (Tanneberger et al., 2022). It is an opportunity for the agribusiness to improve the overall sustainability (Freeman et al., 2022; Liu et al., 2023), and it clearly produces more societal benefits regarding ecosystem services than conventional management (Liu et al., 2023). As regards GHG mitigation, the rise in WTD reduces carbon losses from peat decomposition, but export of carbon in the harvest impairs the carbon balance of the system (Beetz et al., 2013). Emissions of $N_2O$ are generally found to be low in paludiculture (Bianchi et al., 2021), but they can remain high if fertilisers are applied (Bockermann et al., 2024). Emissions of $CH_4$ are affected by the crop type, harvest management and N fertilisation (Boonman et al., 2023), but they can be efficiently reduced by leaving an oxidised, non-waterlogged layer on the peat surface to facilitate microbial oxidation of $CH_4$ (Kandel et al., 2020). Solutions for paludiculture implementation are, for example, forage and willow that can be produced in wet conditions because their roots improve the bearing capacity of the peat and thus ease machine work in wet conditions. Compared to restoration to natural conditions, paludiculture leads to compromises as both ecosystem services and economic productivity are expected to be maintained, and it is not well known how these two aspects are balanced best in practice. Set-aside is often not a planned management option, but wet fields drift to non-productive use when the drainage system degrades, and there are limited data on the GHG balance of such fields.

We established an experimental site with forage, willow and set-aside treatments in wet management on highly decomposed cultivated peat soil in southern Finland in 2019–2023. As the target WTD of 20 cm below the surface was reached only periodically, we cannot call the site a paludiculture site, but the results can be used to discuss the effects and practical issues during the transition period to paludiculture. Our research questions were as follows: (1) what is the carbon and GHG balance of a moderately rewetted drained peatland? (2) How much does harvesting reduce the potential to improve the carbon balance? (3) Do $CH_4$ emissions compromise GHG mitigation in wet management?

## 2 Materials and methods

### 2.1 The site and management

The site was located in southern Finland (60.22° N, 24.78° E, 110 m a.s.l.), and it has been in cultivation at least since the 19th century. The field has been in a crop rotation with cereals and grass during the latest decades. The climate is boreal humid, with long-term (1991–2021) annual mean temperature of 5.2 °C and precipitation of 621 mm (Jokinen et al., 2021). The sum of annual global radiation is 3358 MJ m$^{-2}$ and total sunshine duration 1699 h. Typically, the soil is frozen and has a snow cover from December to March–April. The field was a highly decomposed fen, with peat depth rang-

**Table 1.** Soil properties ($\pm$ standard deviation) in the 0–20 cm layer in 2021.

| Variable | Value |
| --- | --- |
| Decomposition status (von Post) | 8 (7–9) |
| Bulk density (g cm$^{-3}$) | $0.39 \pm 0.05$ |
| Porosity (%) | $0.80 \pm 0.02$ |
| Ash (%) | $42 \pm 3.8$ |
| pH | $5.4 \pm 0.09$ |
| C (g kg$^{-1}$) | $286 \pm 24.6$ |
| N (g kg$^{-1}$) | $15.2 \pm 1.24$ |
| Total P (g kg$^{-1}$) | $0.97 \pm 0.08$ |
| Soluble P (g kg$^{-1}$) | $0.01 \pm 0.001$ |
| K (g kg$^{-1}$) | $0.17 \pm 0.03$ |
| Mn (g kg$^{-1}$) | $0.15 \pm 0.02$ |
| S (g kg$^{-1}$) | $2.01 \pm 0.13$ |
| Al (g kg$^{-1}$) | $1.41 \pm 0.12$ |
| Fe (g kg$^{-1}$) | $5.92 \pm 0.63$ |

ing from 0.8 to over 2 m. Organic carbon content was 25 % and pH 5.5 in the surface layer (0–20 cm) (Table 1). The original subsurface drainage system with tile drains was replaced by modern plastic pipes surrounded by gravel in the 1960s. The distance between the pipes was 18 m until 1979, when it was changed to 9 m. The drainage depth was 60–80 cm, and a control well was installed prior to the experiment to restrict water outflow and raise the groundwater table. The adjustable tube inside the well was set to a position that let the water out when the water table reached 20 cm depth below the soil surface.

The site was established in 2018, and it consists of 12 experimental plots ($9 \times 6$ m) in four blocks (see the key figure). Four replicate plots with either a grass mixture for forage (sown with *Poa trivialis* and *Festuca pratensis*, replanted in 2019 and 2021 with *Phleum pratense*, *Festuca pratensis*, *Lolium multiflorum* and *Poa pratensis*), bog bilberry (*Vaccinium uliginosum*; a.k.a. bog blueberry or bog whortleberry) or willow variety Klara (hybrid of *Salix schwerinii* L. 'Amgunskaja' $\times$ *Salix viminalis* L. 'Ivar') were randomly assigned within the four blocks. The grass was seeded and bilberry seedlings and willow saplings planted in June 2018 (Table S1 in the Supplement). The bilberry did not grow roots, and those plots were left to develop to the set-aside type during the following years; thus, we named this treatment as set-aside. The number of species in all 12 plots was determined once in the summer of 2021.

### 2.2 Ancillary measurements

Biomass growth of willow was monitored by cutting three willow individuals from each plot for determining the aboveground biomass each June. The leaves and stem with branches were separated and weighed to determine fresh biomass. The woody biomass was cut in 10 cm pieces and

dried at 65 °C for 2 weeks. The root biomass around one of the monitored plants per plot was determined by taking $50 \times 80 \times 20$ cm peat samples from three layers: 0–20, 20–40 and 40–60 cm once per year. Visible large ($> 2$ mm) and fine roots were manually separated from the peat, dried and weighed. For determining fine roots, the peat samples were mixed, and a 1 kg subsample was taken. Annual growth in stem, stool and coarse roots was calculated by subtracting the value from the previous year. Annual turnover rate of fine roots was assumed to be 3 times the biomass of fine roots as in Pacaldo et al. (2014). For example, biomass increment in 2019 was calculated with the following equation:

$$\text{annual growth} = F_{19} + (S_{20} - S_{19}) + (St_{20} - St_{19})$$
$$+ (Cr_{20} - Cr_{19}) + 3 \times Fr_{19}, \quad (1)$$

where $F_{19}$ is foliage in 2019, $S_{19}$ and $S_{20}$ are stems in 2019 and 2020, $St_{19}$ and $St_{20}$ are stools in 2019 and 2020, $Cr_{19}$ and $Cr_{20}$ are coarse roots in 2019 and 2020 and $Fr_{19}$ is fine roots in 2019. Subsamples were taken for determining the C content of the dried biomass in 2019 and 2020, and the mean values were used for the following years. The yield per hectare was estimated to be the weight of 25 000 individuals based on 80 cm $\times$ 50 cm spacing.

Soil temperature was measured first at a depth of 10 cm (but at a depth of 5 cm from May 2020 on to achieve better response of $CO_2$ to air temperature) in each treatment with ElcoLog sensors (Elcoplast Oy, Tampere, Finland). The sampling rate was 1 h in summer and 2.5 h in winter. The air temperature, precipitation and radiation data were taken from the Jokioinen weather station of the Finnish Meteorological Institute (FMI, 2024, CC BY 4.0) located about 10 km from the site. Continuous photosynthetically active radiation (PAR) data were produced with global radiation data from FMI and corrected using the ratio of 2.04 for global radiation and PAR (Meek et al., 1984).

WTD was measured from monitoring pipes at the corners of the site at the time of the opaque chamber measurements until 2021, when monitoring pipes were also installed in the centre of each plot. During the summers of 2021 and 2022, there were also HOBO water level data loggers (Onset, Bourne, MA, USA) in each plot for continuous water level monitoring with a sampling rate of 1 h. In winter, when the loggers were not used, WTD was measured manually from monitoring pipes when the water was not frozen.

The leaf area index (LAI) was measured at the same time as the transparent chamber measurements with a portable LAI metre (SunScan; Delta-T Devices Ltd, Cambridge, United Kingdom). LAI values of $> 3$ were set to 3 as they were assumed to not affect photosynthesis due to the saturation of the reflectance (Aparicio et al., 2000). When harvesting the grass plots, the previous measured LAI value was extrapolated to the moment just before harvesting, after which the LAI value was set to 1 as measured. In 2022, we measured green canopy cover with the Canopeo app (Patrignani

and Ochsner, 2015) instead of LAI. Based on our experiences and due to the operation and physical design of the LAI device, it did not provide as comprehensive of a picture of the biomass inside the gas measurement collar as Canopeo. The vegetation index was found to be faster to measure and less dependent on the ambient light conditions than the light interception method (Shepherd et al., 2018). LAI was indexed by dividing by the maximum value of 3 and green canopy cover by 100 (values from 0 % to 100 %) so that the generated vegetation index range was 0–1. The vegetation index was set to 0 from the end of November until mid-April, when snow and frost covered the ground or no green vegetation was present.

Soil samples for analysing the soil properties were taken first in October 2018, and another sampling was conducted in June 2021 with additional analysis. As there were no significant changes in the soil properties between these samplings, we only present the results of the second soil sampling in Table 1. The samples were taken from the 0–20 cm layer using a soil corer with a diameter of 3 cm. Approximately 20 subsamples were pooled to make a composite sample that was air-dried and sieved (2 mm) for the chemical analyses. Soil core samples for dry bulk density and porosity (diameter of 5 cm) were taken from the surface layer (0–17.5 cm) of each plot in October 2020 using the Kopec corer, and the samples were dried at 37 °C for a week. Soil acidity was determined using the ISO 10390 method. Nutrient content was analysed as described in Vuorinen and Mäkitie (1955). Soil carbon and nitrogen were determined using the dry combustion method (Leco TruMac CN; LECO Corporation, MI, USA).

## 2.3 GHG measurements

Dark respiration of the plants together with soil respiration (ecosystem respiration) and fluxes of $N_2O$ and $CH_4$ were measured using opaque chambers biweekly or once per month in the winter between March 2019 and March 2023. In each plot, a 60 cm $\times$ 60 cm steel collar was installed at a depth of 10–15 cm. The location of the collars was 1 m from the short edge of the plot and 3 m from the edges of the adjacent plots. An aluminium chamber (height of 40 cm) mounted at the top of the collar was sealed with water in the groove of the upper edge of the collar. In the winter, NaCl was added to the water to avoid ice formation. The clear aluminium surface effectively reflected light and kept the temperature change moderate inside the chamber. The measurements were done during the daytime between 10:00 EET and 14:00 EET approximately every 2 weeks in summertime and monthly in the winter. The chambers were closed for 30 min, and four 20 mL gas samples were taken with a 60 mL plastic syringe to pre-evacuated vials (Exetainer, Labco Limited, UK) at 10 min intervals starting immediately after closing. Prior to sampling, the syringe was pumped five times to mix the air in the chamber. The samples were analysed with a gas chromatograph (Agilent 7890; Agilent Technologies, Inc.,

Wilmington, DE, USA), equipped with a flame ioniser and electron capture detectors and a nickel catalyst for converting $CO_2$ to $CH_4$. The gas chromatograph had a 2 mL sample loop and a backflush system for separating water from the sample and flushing the precolumn between the runs. The precolumn and analytical columns consisted of 1.8 and 3 m long steel columns, respectively, and were packed with an 80/100 mesh HayeSep Q (Supelco Inc., Bellefonte, PA, USA). Nitrogen was used as the carrier gas, and a standard gas mixture of known concentration of $CO_2$, $N_2O$ and $CH_4$ was used for a calibration curve with seven concentration points. An autosampler (222 XL Liquid Handler; Gilson Medical Electronics, France) fed the samples to the loop of the gas chromatograph.

Net ecosystem exchange (NEE) including photosynthesis and respiration of the soil and plants was measured approximately every 2 weeks during the growing season using a transparent chamber ($60 \times 60 \times 60$ cm) made of polycarbonate plexiglass (1 mm, light transmission of 95 %). The chamber was equipped with a Vaisala GMP-343 probe for $CO_2$ measurement and a temperature and humidity sensor (Vaisala Oy, Vantaa, Finland) and two fans for mixing the air during the measurement. PAR was measured with a LI-190 quantum PAR sensor (LI-COR, Lincoln, NE, USA) inside the chamber. Four measurements with different amounts of light entering were taken from each plot on each measurement day in order to cover a large range of light conditions and facilitate the gap filling by modelling. One or two layers of a white fabric shroud and one blackout curtain were used to acquire measurement results in different light conditions (approximately 100 %, 50 %, 25 % and 0 % of ambient radiation). The measurement with 0 % radiation gave the estimate for ecosystem respiration (ER). The measurements were done in the same collars as the opaque chamber measurements. Each measurement took 1 min with a 5 s sampling rate or 2 min in early or late growing season when the change in flux was minor. The chamber was flushed after each measurement to reconstitute ambient $CO_2$ and air humidity contents. After closing the chamber, a lag time of 10 s was applied to exclude the time when the flux was not yet stabilised. Clear-sky conditions were preferred to avoid problems related to changing cloud cover and to achieve the widest possible range of available light. The temperature change inside the chamber was less than 1.5°, which was also used as a criterion for data filtering.

The change in $CO_2$ concentration during the chamber enclosure was assumed to be linear. The measurement results of $CO_2$ in parts per million (ppm) were converted to $g\,m^{-2}\,h^{-1}$ by the ideal gas law using measured temperature inside the chamber. If the flux was not yet stabilised at the beginning (first four data points) of the measurement, outliers were defined with the MATLAB isoutlier command, resulting in the removal of 210 of total 23 066 data points in 1564 flux measurements.

If the snow cover was thicker than 20 cm, a concentration snow gradient method as in Maljanen et al. (2003) was used to determine the GHG fluxes. A probe made of a steel pipe (Ø 3 mm), with a three-way valve and a plastic syringe, was used to sample 15 mL of air just above the snow cover, at the bottom of the snow cover and at every 10 cm in between in three replicate locations per plot. The gas was stored in the pre-evacuated vials, and the concentrations were determined by gas chromatography.

Measurements for bare soil respiration were made in unvegetated subplots in July 2019–December 2022. For willow, the large $60 \times 60$ cm frames were used, but for forage and set-aside, we installed one sheet-metal air ventilation pipe 27 cm in diameter and 30 cm in length to the depth of 5–10 cm next to the opaque chamber collars in the eight plots of grass and set-aside. All green vegetation within the chamber area was removed, and root growth was limited by cutting around the chamber occasionally with a knife. For the measurements, the cylinders were closed, with a cover equipped with a $CO_2$ sensor (GMP-343; Vaisala Oy, Vantaa, Finland) and a small fan. One measurement lasted for 1 min with a 5 s sampling rate. Measurements were taken about once every week or two and more frequently in summer than in winter. In winters of 2021–2022 and 2022–2023, this method was not used due to too high a snow depth, but measurements with the snow gradient method were utilised (Maljanen et al., 2003).

## 2.4 Flux modelling

Gross photosynthesis (GP) can be determined as the difference between NEE and ER. Instantaneous GP was estimated for each measurement occasion by Eq. (2):

$$GP = NEE - ER, \tag{2}$$

where the full darkened transparent chamber measurement result (ER) is subtracted from the light-dependent flux (NEE) measured during the same day. Thus, we follow the sign convention with positive ER and negative GP values.

The gaps in GP and ER data between the measurement occasions were predicted using hourly time series of the ancillary data. Hourly time points for vegetation index, WTD and soil temperature were acquired from the measured values by linear interpolation. Gaps in soil temperature were filled with the modified soil temperature model (Zheng et al., 1993) using the air temperature. Air temperature and PAR were assumed to be the same for all plots, whereas we used a plot-specific vegetation index and the soil temperature from the certain treatment. Hourly ER and GP were modelled using nonlinear regression (fitnlm function in MATLAB) for all eight plots in forage and set-aside treatments. Empirical models were used for ER as in Lohila et al. (2003) and for GP as in Kandel et al. (2013). Instead of the phytomass indices used in the above publications, we used the vegetation index formed according to the LAI and Canopeo measure-

ments (index from 0 to 1) to describe the stage of the crop growth.

We used the following equation first defined by Long and Hällgren (1993) for GP to estimate empirical coefficients ($A_{max}$ and $k$):

$$GP = \frac{A_{Max} \times PAR}{k + PAR} \times VI \times T_{Scale}, \qquad (3)$$

where PAR is the measured photosynthetically active radiation ($\mu mol\,m^{-2}\,s^{-1}$), VI is the vegetation index, $A_{max}$ is the asymptotic maximum ($g\,CO_2\,m^{-2}\,h^{-1}$), and $k$ is a half-saturation value ($\mu mol\,m^{-2}\,s^{-1}$). $T_{Scale}$ represents the temperature sensitivity of photosynthesis and follows the equation presented by Raich et al. (1991):

$$T_{Scale} = \frac{(T - T_{min})(T - T_{max})}{(T - T_{min})(T - T_{max}) - (T - T_{opt})^2}, \qquad (4)$$

where $T$ is the measured temperature, $T_{min}$ is the photosynthetically active minimum temperature of $-2\,°C$, $T_{max}$ is the maximum of $40\,°C$ and the optimum is $20\,°C$ as in Kandel et al. (2013).

ER was estimated using data from the opaque and fully darkened transparent chambers. The empirical coefficients ($R0_s$, $R0_p$, $E0_s$ and b) were estimated with a nonlinear regression model similarly to the case of GP. Annual fluxes were computed as sum of the hourly fluxes with a trapezoidal method (trapz function in MATLAB v2019b). ER consists of autotrophic respiration ($R_{auto}$), i.e. plant respiration, and heterotrophic respiration ($R_{hetero}$), i.e. soil respiration (Lloyd and Taylor, 1994), with the extension of WTD as in Karki et al. (2014):

$$ER = R_{hetero} + R_{auto}, \qquad (5)$$

$$R_{hetero} = R0_s \times \exp\left(E0_s\left(\frac{1}{56.02} - \frac{1}{T_{soil} + 46.02}\right)\right)$$
$$+ b \times WTD, \qquad (6)$$

$$R_{auto} = VI \times R0_p \times \exp\left(b_d\left(\frac{1}{10 + 273} - \frac{1}{T_{air} + 273}\right)\right), \qquad (7)$$

where $T_{soil}$ is the measured soil temperature, VI is the vegetation index, $T_{air}$ is the measured air temperature, $R0_s$ is soil respiration at the reference temperature of $10\,°C$ ($g\,CO_2\,m^{-2}\,h^{-1}$), $R0_p$ is plant respiration at the reference temperature at $10\,°C$ ($g\,CO_2\,m^{-2}\,h^{-1}$), $b$ is the effect of WTD, $E0_s$ is ecosystem sensitivity and $b_d$ is the temperature dependence of dark respiration set to 5000 as in Lohila et al. (2003). Bare soil respiration was estimated like ER but using only Eq. (6). The estimated parameters $R0s$, $Es$ and $b$ and model $R^2$ are shown in Table S3.

## 2.5 Data processing and analysis

For the transparent chamber measurements, the criterion $R^2 > 0.9$ for the fitted linear assumption of flux measurements would exclude a large amount of data, especially with a small change in $CO_2$, leading to a biased dataset. Therefore, we decided to add the criterion $S_{xy} < 2.3\,g\,CO_2\,m^{-2}\,h^{-1}$ for the dataset like in Kutzbach et al. (2007) ($S_{xy}$ is the standard deviation of the residuals, and $2.3\,g\,m^{-2}\,h^{-1}$ is the 95th percentile of measurements). This procedure resulted in the removal of 59 values out of total 1467 measurements. In the modelling phase, fitted values were examined, and outliers were removed to avoid distortion. Outliers were defined as observations with an absolute value of standardised residuals greater than 3. In 2019, 3 out of 260 GP values and 3 out of 243 ER values were removed. In 2020, none of the 200 GP values and 2 of 230 ER values were removed. Overall, 2 out of 365 GP values and 4 out of 247 ER values were removed in 2021, and 12 out of 583 GP values and 2 out of 323 ER values were removed in 2022. The model's estimated parameters $A_{max}$, $k$ of GP, $R0_s$, $R0_p$, $E_s$ and $b$ of ER and model correlations are shown in Table S2. The measured versus model-predicted values of GP and ER are shown by treatments and years in Fig. S1 in the Supplement.

For bare soil respiration measurements in set-aside and forage, the same criteria were used as for the transparent chamber ($R^2 > 0.9$ and $S_{xy} < 95\,\%$), leading to a removal of 12 values of the total 601. In bare soil measurements in mid-summer 2022, 24 flux measurements occurred (nine values in one plot and none–five in others) of the total 147 values, which were inexplicably high ($3-18\,g\,CO_2\,m^{-2}\,h^{-1}$). Values were of the same magnitude as values measured immediately after ploughing in Honkanen et al. (2023). We decided to remove these values as outliers to avoid model distortion. Soil respiration of willow was defined with the opaque chamber method, and such outliers did not occur in these measurements. In the modelling phase, outliers defined as observations with an absolute value of standardised residuals greater than 3 were removed, resulting in the removal of 13 measurements of the total 984 measurements (including all plots).

A linear regression model was fitted to calculate gas concentrations, and the ideal gas law was used to solve the flux rate for every enclosure of the opaque chambers. Nonlinear responses of $CO_2$ indicated a leaking chamber or other problem in the measurement, and thus, if the $R^2$ of $CO_2$ was less than 0.9, the results of $CH_4$ and $N_2O$ were also discarded. In addition, sudden variations in $CH_4$ fluxes due to ebullition were filtered by selecting only flux rates with an intercept between 1.5 and 2.4 ppm. These criteria resulted to 176, 117 and 118 discarded values out of 1044 in the case of $CH_4$, $CO_2$ and $N_2O$, respectively. All data cleaning and processing was done with MATLAB (v2019b; The Math Works, Inc.).

## 2.6 GHG balance

The annual net ecosystem carbon balance was constructed as the sum of the hourly values of NEE and yield data for each year in the case of forage and set-aside treatments. Modelling was used to fill the gaps between the measurement occasions to create a continuous series of hourly values. For willow, annual estimates of carbon loss in soil respiration were available from the chamber measurements from the unvegetated frames, but as the willows were too high for the chambers, their net production had to be estimated based on biomass accumulation during 4 years (Pacaldo et al., 2014). The presented net ecosystem carbon balance of willow is thus the sum of average annual $CO_2$-C from soil respiration and average annual amount of carbon bound in the biomass during the first 4 cultivation years. The cumulative annual fluxes of $CH_4$ and $N_2O$ for each management practice were calculated by interpolating the emissions between consecutive sampling days. Global warming potentials 27 and 273 were used for $CH_4$ and $N_2O$, respectively, to convert the results to $CO_2$ equivalent for the total GHG balance (Forster et al., 2021).

## 2.7 Statistical analyses

Linear mixed models were used to find variables explaining variation in the gas fluxes. Crop, year, WTD and all their interactions were denoted as fixed effects. Block and block–year interaction were assumed to be independent and normally distributed random effects. The most suitable covariance structure was chosen using Akaike's information criterion (AIC). The models were fitted using the residual maximum likelihood (REML) method, and degrees of freedom were estimated using the Kenward–Roger method. The residuals were plotted against the fitted values, and the normality of the residuals was checked using box plots. The data were log-transformed when needed to normalise the distributions. The method of Tukey–Kramer was used for all pairwise comparisons of means with a significance level of 0.05. After the first model run with all relevant variables, the non-significant variables were removed one by one to find the most relevant effects. All statistical analyses were performed using the SAS Enterprise Guide v7.1 (SAS Institute Inc., Cary, NC, USA).

## 3 Results

### 3.1 Climate and site variables

Annual mean temperature was 6.9, 6.0, 5.8 and 5.8 °C and annual precipitation 750, 600, 660 and 546 mm in 2019–2022, respectively. The number of days with snow cover on the soil within each modelling year (April to March) was 13, 81, 108 and 118, respectively. The annual mean temperature during the study years was higher than the long-term average of 5.2 °C in 1991–2020 (Jokinen et al., 2021). Two

**Table 2.** Four-year cumulative carbon balance of willow ($\pm$ standard deviation). Negative sign indicates sequestered carbon and positive sign released carbon to the atmosphere.

| Component | Mg C ha$^{-1}$ 4 yr$^{-1}$ | % of total |
|---|---|---|
| Stem (harvested) | $-50.7 \pm 14.3$ | 59 |
| Foliage | $-6.2 \pm 0.8$ | 7 |
| Aboveground stool | $-12.6 \pm 3.2$ | 15 |
| Underground stool | $-8.5 \pm 3.8$ | 10 |
| Coarse roots | $-3.9 \pm 1.5$ | 4 |
| Fine roots | $-4.6 \pm 0.6$ | 5 |
| Total sequestered carbon | $-86.5 \pm 19.5$ | |
| Soil respiration | $43.5 \pm 2.7$ | |
| Net ecosystem exchange | $-43.1 \pm 21.1$ | |
| Net ecosystem carbon balance | $7.6 \pm 7.7$ | |

study years exhibited lower and two higher annual precipitation as compared to the long-term mean of 621 mm. The WTD showed an increasing trend in time and high within-year variation (Fig. 1). The average WTD was $-54$, $-41$, $-39$ and $-27$ cm in 2019–2022, respectively. WTD varied from $-89$ to $-4$, $-77$ to 2, $-120$ to 1.4 and $-100$ to 1.8 cm in 2019–2022, respectively.

The forage yields were $6.3 \pm 0.9$, $8.9 \pm 0.7$, $11 \pm 0.8$ and $9.4 \pm 0.9$ Mg dry matter ha$^{-1}$ in 2019–2022, respectively. There were two harvests in 2020 and three in the other years. The plots were dominated by *Phleum pratense* and *Festuca pratensis* in 2021. The dry mass yields of willow were $30 \pm 14$ and $73 \pm 28$ Mg dry matter ha$^{-1}$ in the harvests of February 2021 and 2023. Most of the C accumulation occurred in the stem (59 %) followed by stool (25 %), roots (9 %) and foliage (7 %) (Table 2). Vegetation of the set-aside plots in 2021 was dominated by wild plants belonging to Families Asteraceae, Cichoriaceae and Caryophyllaceae. Bog bilberry covered 1 % or less on each of the four replicate plots. The set-aside vegetation had the highest species diversity involving 19 vascular plants compared to 12 at willow plots and 9 at forage plots, with the latter two including crop plants.

### 3.2 Carbon balance

Model-predicted maximum hourly GP was $-0.7$, $-3.2$, $-4.3$ and $-4.8$ g $CO_2$ m$^{-2}$ h$^{-1}$ in the set-aside plots in 2019–2020, and $-3.9$, $-5.9$, $-4.3$ and $-4.8$ g $CO_2$ m$^{-2}$ h$^{-1}$ in the forage plots in 2019–2022, respectively (Fig. S1). The maximum measured GP value was $-1.1$, $-2.4$, $-3.4$ and $-4.5$ g $CO_2$ m$^{-2}$ h$^{-1}$ for set-aside and $-3.4$, $-6.2$, $-4.8$ and $-4.2$ g $CO_2$ m$^{-2}$ h$^{-1}$ for forage in 2019–2020, respectively. Annual values of GP varied from $-9.3$ to $-12$ Mg $CO_2$-C ha$^{-1}$ yr$^{-1}$ in the forage and from $-1.5$ to $-10$ Mg $CO_2$-C ha$^{-1}$ yr$^{-1}$ in the set-aside treatment (Table 3). The variables initially included in the analysis were annual mean

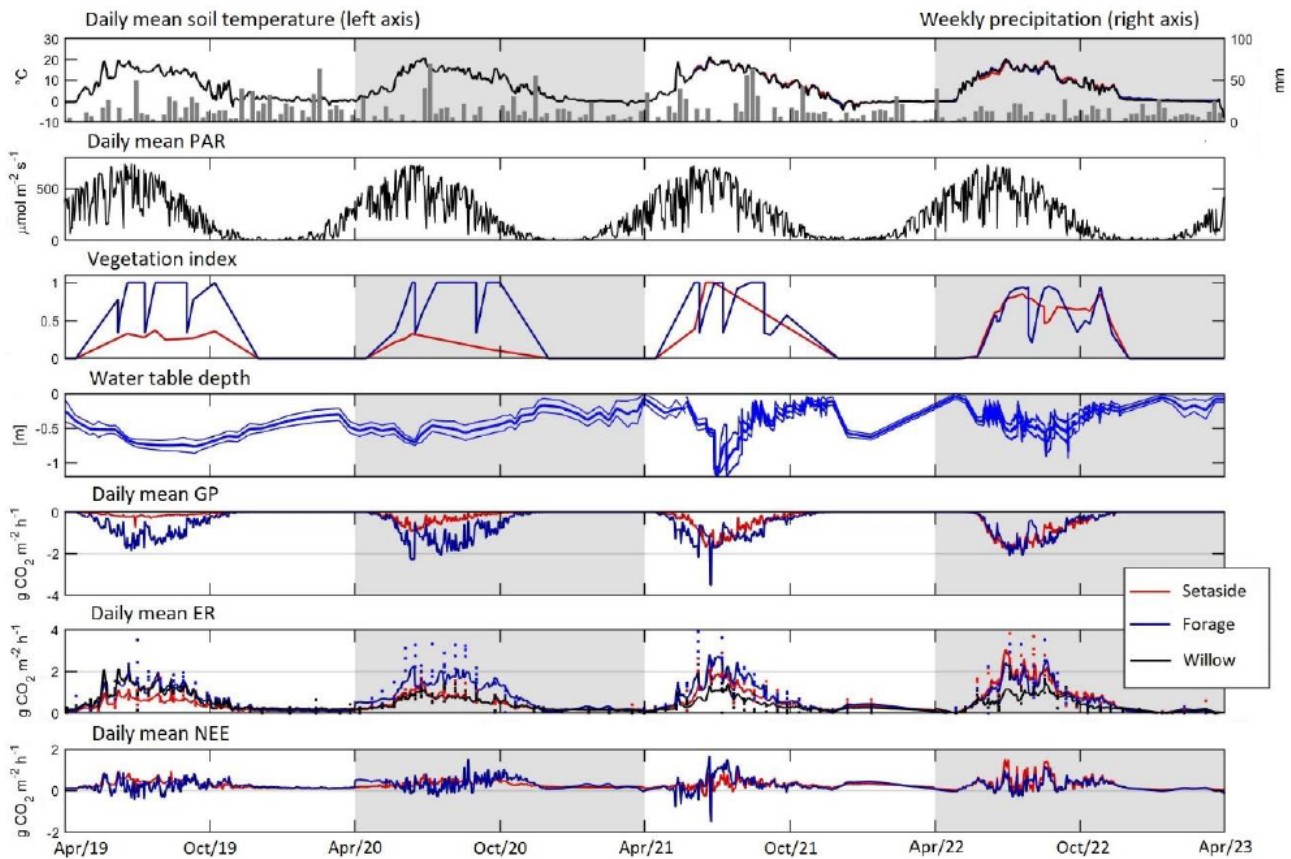

**Figure 1.** Daily mean of soil temperature and precipitation, photosynthetically active radiation (PAR), vegetation index, water table depth (site mean ± SD), gross photosynthesis, measured (dots) and model-predicted (line) ecosystem respiration (soil respiration for willow), and net ecosystem exchange. Annual modelling periods (April–March) are marked by a light-grey or white background.

WTD, crop type and year as the main effects and all their interactions. Finally, only crop, year and their interaction were left in the model (Table S4). Inclusion of WTD did not result in a meaningful estimate as both the productivity of especially the set-aside established in 2019 and the WTD increased with years, and the observed change in GP did not, in reality, increase with WTD but with time. Forage and set-aside treatments differed significantly ($p < 0.001$), there was an increasing trend in productivity from 2019–2022 ($p = 0.0009$), and the differences between the crop types were the highest in 2019–2021 ($p < 0.001$).

Modelled maximum hourly ER was 2.4, 2.3, 3.0 and 4.7 g $CO_2$ m$^{-2}$ h$^{-1}$ in the set-aside and 3.5, 3.4, 4.8 and 4.5 g $CO_2$-C m$^{-2}$ h$^{-1}$ in the forage plots (Fig. S1). Measured maximum ER with the opaque chamber method was 1.7, 2.0, 2.9 and 4.6 g $CO_2$-C m$^{-2}$ h$^{-1}$ for the set-aside and 3.5, 4.2, 4.0 and 4.8 g $CO_2$-C m$^{-2}$ h$^{-1}$ for forage plots. Annual ER varied from 14 to 19 Mg $CO_2$-C ha$^{-1}$ yr$^{-1}$ in the forage and from 8 to 17 Mg $CO_2$-C ha$^{-1}$ yr$^{-1}$ in the set-aside treatment (Table 3). Variables initially included in the analysis were annual mean WTD, crop type and year as the main effects as well as their interactions. WTD did not explain the variation

in the annual ER estimate well for likely the same reason as for GP since plant respiration is related to the biomass of the vegetation, which increased during the experimental years. The best model was based on the crop type and year as the main effects and their interactions (Table S4). In this model, crop type explained ER well, ER increased in time and ER increased between years more in forage than set-aside.

Hourly model-predicted NEE varied from $-2.9$ to 3.7 g $CO_2$ m$^{-2}$ h$^{-1}$ in the set-aside and from $-4.6$ to 3.7 g $CO_2$ m$^{-2}$ h$^{-1}$ in the forage treatment (results not shown). There were 29, 28, 16 and 47 days annually with negative daily NEE in the forage plots during the study years, respectively, and fewer such days (1, 0, 9 and 6) in the set-aside plots in 2019–2022 (Fig. 1). The cumulative annual balance ranged from 5.1 to 8.0 Mg $CO_2$-C ha$^{-1}$ yr$^{-1}$ in the forage and from 6.8 to 7.1 Mg $CO_2$-C ha$^{-1}$ yr$^{-1}$ in the set-aside treatment (Table 3), and the treatments did not statistically differ. The net ecosystem carbon balance (NECB) that accounts the amount of carbon exported in the harvested yield varied from 5.5 to 13.5 Mg $CO_2$-C ha$^{-1}$ yr$^{-1}$ in the forage treatment and was equal to NEE in the set-aside treatment (Table 3). The NECB values differed statistically between the

**Table 3.** The estimated annual sums ($\pm$ standard deviation) of gross photosynthesis (GP), ecosystem respiration (ER), net ecosystem exchange (NEE), carbon exported in the harvested yield, net ecosystem carbon balance (NECB), $N_2O$ and $CH_4$ effluxes and the total emissions (global warming potential of 100 years; GWP-100), with either NEE or NECB representing $CO_2$ emissions in the forage and set-aside plots, and selected data for willow. Significant differences ($p < 0.05$) between treatments within a year are denoted by different letters ($n = 4$).

| Year | Variable and unit | Forage | Set-aside | Willow[a] |
|---|---|---|---|---|
| 2019 | GP Mg $CO_2$-C ha$^{-1}$ | $-9.31 \pm 0.75$[a] | $-1.48 \pm 0.32$[b] | |
| | ER Mg $CO_2$-C ha$^{-1}$ | $14.4 \pm 2.17$[a] | $8.25 \pm 2.40$[b] | |
| | NEE Mg $CO_2$-C ha$^{-1}$ | $5.08 \pm 1.80$ | $6.77 \pm 2.41$ | |
| | C in yield Mg C ha$^{-1}$ | $3.17 \pm 0.4$ | $0$ | |
| | NECB Mg C ha$^{-1}$ | $8.25 \pm 2.13$ | $6.77 \pm 2.41$ | |
| | Soil respiration Mg $CO_2$-C ha$^{-1}$ | $12.8 \pm 4.99$ | $11.4 \pm 1.82$ | $14.8 \pm 0.76$ |
| | $N_2O$-N kg ha$^{-1}$ | $11.9 \pm 7.60$[a] | $32.6 \pm 12.1$[b] | $17.4 \pm 10.3$ |
| | $CH_4$-C kg ha$^{-1}$ | $-0.28 \pm 0.75$ | $-1.00 \pm 0.73$ | $-1.64 \pm 0.26$ |
| | $GWP_{100}$ Mg $CO_2$ eq. ha$^{-1}$ (NEE)[b] | $23.7 \pm 5.67$ | $38.8 \pm 19.8$ | |
| | $GWP_{100}$ Mg $CO_2$ eq. ha$^{-1}$ (NECB)[c] | $35.3 \pm 6.86$ | $38.8 \pm 19.8$ | |
| 2020 | GP Mg $CO_2$-C ha$^{-1}$ | $-11.7 \pm 1.17$[a] | $-3.57 \pm 0.46$[b] | |
| | ER Mg $CO_2$-C ha$^{-1}$ | $19.3 \pm 1.85$[a] | $10.6 \pm 1.49$[b] | |
| | NEE Mg $CO_2$-C ha$^{-1}$ | $7.64 \pm 1.75$ | $7.05 \pm 1.12$ | |
| | C in yield Mg C ha$^{-1}$ | $3.35 \pm 1.28$ | $0$ | |
| | NECB Mg C ha$^{-1}$ | $11.0 \pm 2.02$ | $7.05 \pm 1.12$ | |
| | Soil respiration Mg $CO_2$-C ha$^{-1}$ | $9.09 \pm 4.86$ | $10.6 \pm 0.97$ | $10.0 \pm 0.79$ |
| | $N_2O$-N kg ha$^{-1}$ | $6.26 \pm 3.39$ | $6.59 \pm 2.97$ | $4.61 \pm 2.99$ |
| | $CH_4$-C kg ha$^{-1}$ | $-0.36 \pm 0.40$ | $-1.01 \pm 0.56$ | $-1.13 \pm 0.33$ |
| | $GWP_{100}$ Mg $CO_2$ eq. ha$^{-1}$ (NEE) | $30.4 \pm 7.64$ | $28.7 \pm 3.73$ | |
| | $GWP_{100}$ Mg $CO_2$ eq. ha$^{-1}$ (NEBC) | $42.6 \pm 8.69$ | $28.7 \pm 3.73$ | |
| 2021 | GP Mg $CO_2$-C ha$^{-1}$ | $-9.46 \pm 1.20$[a] | $-6.34 \pm 0.68$[b] | |
| | ER Mg $CO_2$-C ha$^{-1}$ | $17.4 \pm 1.40$[a] | $13.5 \pm 1.82$[b] | |
| | NEE Mg $CO_2$-C ha$^{-1}$ | $7.95 \pm 2.16$ | $7.12 \pm 2.16$ | |
| | C in yield Mg C ha$^{-1}$ | $5.54 \pm 0.46$ | $0$ | |
| | NECB Mg C ha$^{-1}$ | $13.5 \pm 1.88$[a] | $7.12 \pm 2.16$[b] | |
| | Soil respiration Mg $CO_2$-C ha$^{-1}$ | $7.82 \pm 2.30$ | $11.5 \pm 2.44$ | $8.99 \pm 2.70$ |
| | $N_2O$-N kg ha$^{-1}$ | $6.49 \pm 3.88$[a] | $2.18 \pm 0.24$[b] | $5.75 \pm 6.25$ |
| | $CH_4$-C kg ha$^{-1}$ | $7.92 \pm 12.7$ | $0.58 \pm 1.87$ | $3.89 \pm 6.12$ |
| | $GWP_{100}$ Mg $CO_2$ eq. ha$^{-1}$ (NEE) | $32.2 \pm 8.16$ | $27.1 \pm 8.03$ | |
| | $GWP_{100}$ Mg $CO_2$ eq. ha$^{-1}$ (NEBC) | $52.2 \pm 6.93$ | $27.1 \pm 8.03$ | |
| 2022 | GP Mg $CO_2$-C ha$^{-1}$ | $-9.34 \pm 2.13$ | $-9.65 \pm 2.08$ | |
| | ER Mg $CO_2$-C ha$^{-1}$ | $14.4 \pm 3.29$ | $16.5 \pm 3.21$ | |
| | NEE Mg $CO_2$-C ha$^{-1}$ | $5.10 \pm 1.15$ | $6.82 \pm 1.15$ | |
| | C in yield Mg C ha$^{-1}$ | $4.72 \pm 0.50$ | $0$ | |
| | NECB Mg C ha$^{-1}$ | $5.46 \pm 6.37$ | $6.82 \pm 1.15$ | |
| | Soil respiration Mg $CO_2$-C ha$^{-1}$ | $8.40 \pm 1.48$ | $15.0 \pm 5.32$ | $9.70 \pm 2.3$ |
| | $N_2O$-N kg ha$^{-1}$ | $9.54 \pm 4.49$[a] | $3.07 \pm 0.86$[b] | $1.69 \pm 1.10$[b] |
| | $CH_4$- kg ha$^{-1}$ | $7.74 \pm 0.59$ | $11.3 \pm 7.72$ | $10.9 \pm 12.9$ |
| | $GWP_{100}$ Mg $CO_2$ eq. ha$^{-1}$ (NEE) | $22.9 \pm 3.34$ | $26.6 \pm 4.61$ | |
| | $GWP_{100}$ Mg $CO_2$ eq. ha$^{-1}$ (NEBC) | $43.3 \pm 3.18$ | $26.6 \pm 4.61$ | |

[a] All components of the carbon balance are not available for willow; see Sect. 2.6. [b] NEE representing $CO_2$. [c] NEBC representing $CO_2$.

forage and set-aside treatments across all years ($p > 0.001$), and inclusion of additional effects in the analysis did not improve the model (Table S4).

Annual sum of respiration varied from 8 to 15 Mg $CO_2$-C ha$^{-1}$ yr$^{-1}$ in the different treatments and years (Table 3). The proportion of soil respiration of the total ecosystem respiration varied from 45 to 90 % in the forage plots and from

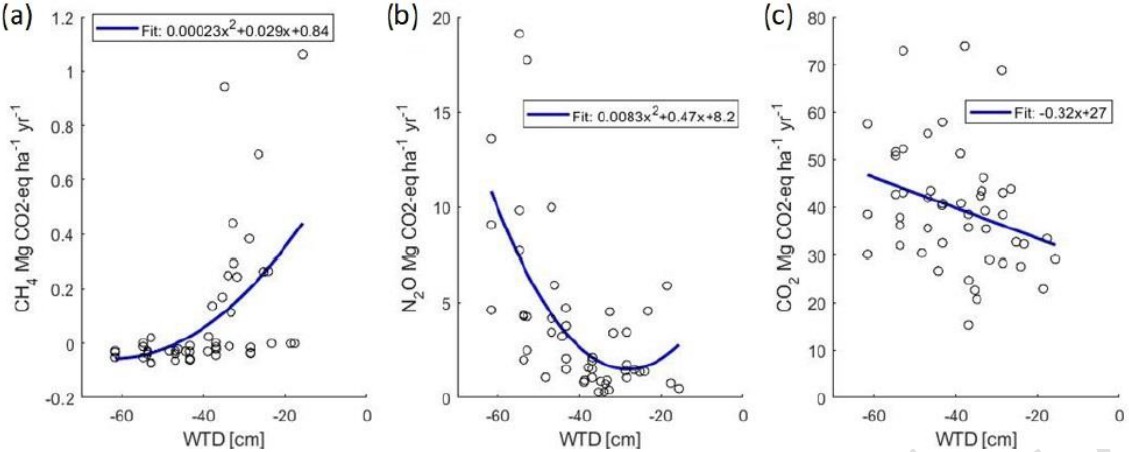

**Figure 2.** Plot-wise mean annual fluxes of $CH_4$ **(a)**, $N_2O$ **(b)** and soil respiration **(c)** ($CO_2$ eq.) as related to the mean annual WTD.

85 % to 100 % in the set-aside plots in 2019–2022. In the set-aside plots, estimated annual bare soil respiration exceeded the estimated ER in all plots in 2019, two plots in 2020 and one plot in 2021 and 2022, and those values were not used in the above calculation, and thus it is assumed that total respiration constituted of only soil respiration in 2019. Annual cumulative soil respiration was explained by WTD (Fig. 2; $p = 0.053$) and crop type ($p = 0.033$), so forage and set-aside treatments were significantly different in the whole dataset and respiration increased in the order forage < willow < set-aside (Fig. S4; Table S4). Plots of the bare soil respiration in relation to WTD and temperature show that there is a clear trend of decreasing respiration with rising WTD (Fig. S2). Three individual curves indicate a contrasting trend, but these three estimations are based on a small number of measurement results. Based on all annual estimates of soil respiration, a 0.1 m rise in WTD reduces respiration by $0.87\,\mathrm{Mg\,CO_2}$-$\mathrm{C\,ha^{-1}\,yr^{-1}}$.

The cumulative total amount of C in the above- and belowground willow biomass was $86.5\,\mathrm{Mg\,C\,ha^{-1}}$ during the 4 study years (Table 2). About 40 % of the carbon in the biomass was left at the site after harvest, and soil respiration amounted to $43.5\,\mathrm{Mg\,ha^{-1}}$, leading to a strongly negative cumulative NEE of $-43\,\mathrm{Mg\,ha^{-1}}$. Carbon export in the harvest changed the net balance to a net loss of 7.6 Mg, corresponding to an average annual $CO_2$ rate of 7 Mg of $CO_2$.

### 3.3 $CH_4$ fluxes

Hourly fluxes of $CH_4$ varied between $-50$ and $30\,\mu\mathrm{g\,CH_4}$-$\mathrm{C\,m^{-2}\,h^{-1}}$ during the first half of the experimental period (Fig. 3). In conjunction with the rise in the WTD, the values of hourly fluxes increased and varied between $-40$ and $900\,\mu\mathrm{g\,m^{-2}\,h^{-1}}$ during the latter half of the period. The annual flux of $CH_4$ varied from $-1.6$ to $11\,\mathrm{kg\,CH_4}$-$\mathrm{C\,ha^{-1}\,yr^{-1}}$, with an increasing trend towards the end of the measurement period (Table 3; Fig. S3). When the mean annual WTD was below 40 cm, the soil was mainly consuming $CH_4$, but the consumption tended to change to emissions as the WTD rose (Fig. 2). Variation in the annual cumulative fluxes of each plot was explained by the WTD ($p = 0.015$) but not by crop type (Table S4). The increasing trend between the years 2019 and 2022 was also shown in the mixed-model analysis as the year had a significant effect ($p = 0.0003$) and the effect of WTD decreased with time (years).

### 3.4 $N_2O$ fluxes

Hourly fluxes of $N_2O$ varied between $-3$ and $2500\,\mu\mathrm{g\,N_2O}$-$\mathrm{N\,m^{-2}\,h^{-1}}$ during the 4 years, with the highest emissions during the first 4 months (Fig. 3). Annual fluxes of $N_2O$ varied from 1.7 to $33\,\mathrm{kg\,N_2O}$-$\mathrm{N\,ha^{-1}\,yr^{-1}}$ (Table 3). The emissions declined in time (Fig. S3), especially in the case of set-aside and willow, whereas those of forage did not show such a trend (Table 3). WTD explained the variation in $N_2O$ fluxes well ($p = 0.015$) (Fig. 2; Table S4). There were some interactions between the year and the crop, but the crop type did not affect $N_2O$ emissions systematically between years. Annual $N_2O$ fluxes of the forage and willow treatments differed in the whole time series ($p = 0.026$).

### 3.5 Global warming potential

The total emissions expressed as $CO_2$ equivalent ranged from 23 to $39\,\mathrm{Mg\,ha^{-1}\,yr^{-1}}$ with NEE as the $CO_2$ component and from 27 to 52 Mg with the C export in harvest taken into account in the forage and set-aside treatments (Table 3). For willow, the annual NECB cannot be calculated, but based on the 4-year estimate on carbon binding in the biomass and carbon exported in the harvest divided to a single year, together with the average annual soil respiration and $N_2O$ and $CH_4$ fluxes, the average annual climate impact of willow cultivation was $10.2\,\mathrm{Mg\,ha^{-1}\,yr^{-1}}$.

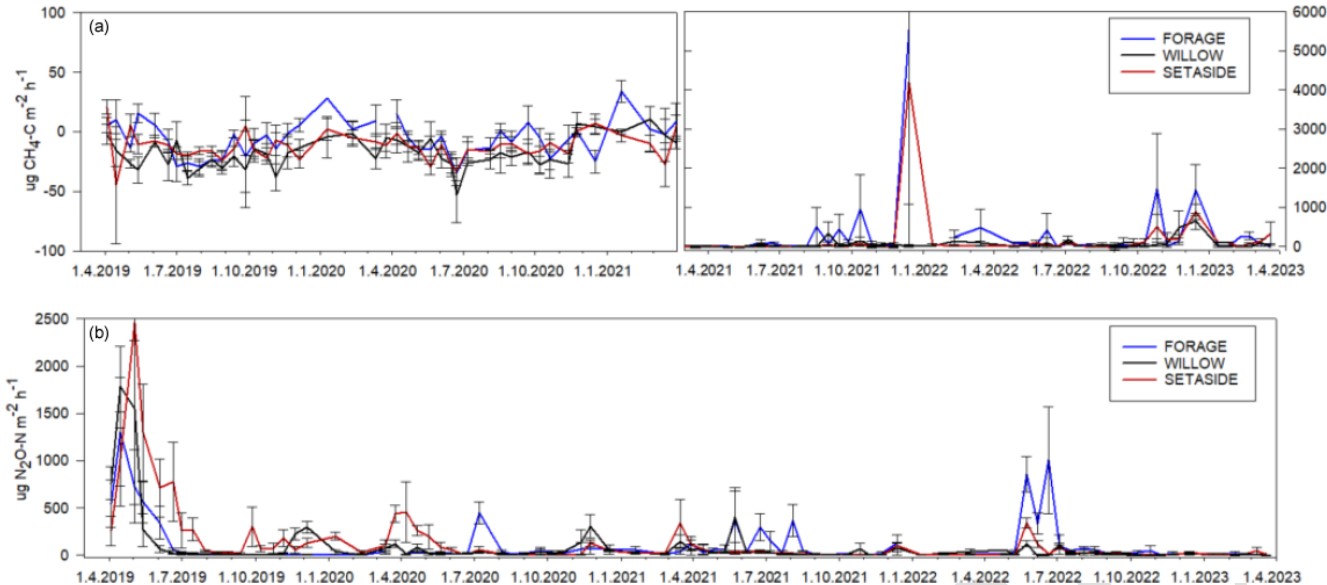

**Figure 3.** Fluxes of $CH_4$ **(a)** and $N_2O$ **(b)** in 2019–2023. The error bars denote standard error. Note the different scale in the $y$ axis in panel **(a)** for the latter half of the period.

## 4   Discussion

NEE values of 5–8 Mg C ha$^{-1}$ yr$^{-1}$ in the forage plots were of the same magnitude as values reported for grass cultivation in northern Europe (Maljanen et al., 2010). They were,
[5] however, 6–10 times higher than NEE reported for the year 2002 in a nearby field (Lohila et al., 2004), highlighting the spatial and temporal variation in soil emissions. During the first 2 years of the experiment, the $CH_4$ fluxes of the forage plots were negative, indicating net consumption of $CH_4$ by
[10] the soil microorganism. The $CH_4$ oxidation rates were generally higher than average values reported from Nordic cultivated peat soils, which have shown net positive values for grass fields (Maljanen et al., 2007, 2010). There was a change from negative fluxes of $CH_4$ to relatively high emissions af-
[15] ter the annual mean water table rose above $-40$ cm during the latter 2 years of the experiment. However, compared to rewetted agricultural sites in the temperate zone, the values of ca. 8 kg $CH_4$-C ha$^{-1}$ were clearly lower than the average of 180 kg $CH_4$-C ha$^{-1}$ yr$^{-1}$ found in temperate paludiculture-
[20] like grassland ecosystems (Bianchi et al., 2021). The $N_2O$ emissions ranging from 6 to 12 kg N ha$^{-1}$ annually were typical of northern European grass fields on organic soils as they were within the 95 % confidence interval of the reported values from temperate and boreal regions (Hiraishi et al., 2014).
[25] After the high emission peak in the beginning of the experiment, there were only short-term peaks after fertilisation. One of them was especially high and long-lasting and likely induced by heavy rainfall after a long dry period, which coincided with fertilisation in May–June 2022. It is typical that
[30] high peaks after fertilisation occur when fertilisation is followed by rainfall (Dobbie et al., 1999), and fertiliser-induced

peaks may be totally absent if there is no coinciding rainfall (Beetz et al., 2013). The set-aside plots with slowly evolving vegetation had clearly lower GP than the forage plots during the first 3 years. However, the ER was also lower [35] in the set-aside treatment, and the resulting NEE was of the same magnitude in both treatments. Because there was no biomass export from the set-aside treatment, the NECB was lower than in the forage treatment in most years. The modelled NEE values were about double compared to long-term [40] abandoned croplands in the Nordic countries (Maljanen et al., 2010), but in our study, the plots did not represent similar ecosystems as they were "abandoned" only for a short period. $N_2O$ fluxes of the set-aside plots were extremely high in 2019 compared to results of previous studies on cultivated [45] peat soils in Nordic countries (Maljanen et al., 2010). As the set-aside was fertilised and unsuccessfully planted with bog bilberry, the high emissions were likely due to abundant free mineral nitrogen in the absence of plant nutrient uptake. As the berry plants did not thrive, the soil was bare for a [50] long period, and the $N_2O$ emissions remained higher than in the other treatments throughout the summer. Such conditions also prevailed in a similar bare fallow treatment at a nearby site in 2000–2002, yielding average $N_2O$ emissions of 25 kg N ha$^{-1}$ yr$^{-1}$ (Regina et al., 2004). During the sec- [55] ond year, the emissions lowered, but as the plots were also fertilised in 2020, they still exhibited emissions as high as in the forage plots. During the last 2 years, the $N_2O$ emissions were at a notably low level, which likely resulted from ceasing fertilisation and a slightly higher WTD leading to less [60] peat being exposed to aerobic conditions. Raising the WTD has been found to diminish $N_2O$ emissions in several studies (van Beek et al., 2010; Leppelt et al., 2014).

Willow grew well at this site, and the mean annual yields were in the higher end of the range of $4\text{–}16\,\mathrm{Mg\,ha^{-1}}$ estimated for northern climate conditions (Viherä-Aarnio et al., 2022). The amount of carbon lost in soil respiration was lower than the amount sequestered in the willow biomass in all years except in 2019, leading to a highly negative NEE during the whole rotation. However, the amount of carbon exported in harvest exceeded the NEE, and the yielded NECB indicated a net loss of carbon to the atmosphere. Although the average annual NECB calculated from the 4-year carbon balance ($1.9\,\mathrm{Mg\,C\,ha^{-1}\,yr^{-1}}$) was low compared to the forage or set-aside treatments, it still indicated a climate-warming end result in short-rotation cropping of willow on peat soil. It is possible to achieve a net positive NEBC in willow cultivation on mineral soils (Harris et al., 2017; Morrison et al., 2019), but in peatlands, the high rate of soil respiration inevitably reduces this potential (Kasimir et al., 2018).

The target WTD was not reached for most of the time likely because there was unexpected lateral water outflow from the site. Our strategy of raising the WTD at a limited area within a field parcel was thus not successful. The larger the area where the water outflow is restricted, likely the better the result, and catchment level water management planning is often recommended for achieving the best results (Mitsch and Wilson, 1996; Pasquet et al., 2015).

The annual emissions can be compared to a well-drained cereal site (oats; mean WTD of 68 cm) on the same field (Honkanen et al., 2023) as the distance between these two experiments was just about 20 m. In 2020, when similar measurements were conducted in both experiments, the total GHG balance (GWP100 with harvest) was 29 Mg in the set-aside and 43 Mg in the forage treatment, while it was $39\,\mathrm{Mg\,CO_2}$ eq. in the conventionally managed cereal plots. As the comparable number for willow was 10 Mg in the willow treatment, it can be argued that the set-aside treatment and willow cultivation with a moderate rise of WTD were better management options than cultivation of annual crops with a typical drainage depth. It was also clear that willow had the best GHG balance of these three management options, which is in agreement with findings of grassland and willow cropping in southern England (Harris et al., 2017). However, the total emissions were still relatively high, suggesting that this kind of moderately wet management is not an efficient climate mitigation measure. This was also shown by the modelling results of Kasimir et al. (2018), concluding that fully rewetted peatland had the most favourable carbon balance and less emissions from soil in a comparison of four different peatland management scenarios. However, management decisions, such as cutting height, also play a role in determining the final carbon balance in short-rotation cropping (Berhongaray et al., 2017).

Set-aside is a relevant management option to study because many cultivated peat fields end up as uncultivated plots when their drainage system degrades and the landowner finds them too wet for cultivation. The annual total emissions were lower in the set-aside plots compared to the forage plots in 2020 and 2021 and also in 2022 if the carbon exported in harvest is taken into account. However, they were not especially low as compared to cultivated peat soils in general. Thus, leaving cultivated peat soils uncultivated without active rewetting is not a desirable form of land management as these sites drift out from food production, but the GHG emissions can remain high. A recent Swedish study also found that setting aside did not reduce GHG emissions from a drained peat soil (Keck et al., 2024)

Our set-aside plots were actually intended to be vegetated by bog bilberry, a native mire plant that could become a novel antioxidant-rich ingredient for food (Lätti et al., 2010) or pharmaceutical (Esposito et al., 2019) industries. However, we soon noticed that the seedlings did not grow roots, indicating that formerly agriculturally cultivated peat was not a suitable substrate for this plant. As the nutrient content of the topsoil did not show large deviations from the reported ranges supporting the growth of bog bilberry (Jacquemart, 1996), it is likely that pH 5.4 at our site was too high. Bog bilberry is usually found in soils with pH below 5. However, in recent trials, it has been successfully grown on Chinese farmlands with pH 5–6, but low pH also improved the growth there (Duan et al., 2022).

There are usually high uncertainties in the GHG measurements, and this is especially true regarding the combination of methods chosen for the willow treatment. The carbon balance of willow was determined using a combination of the pool-based and flux-based methods, which can differ by several magnitudes (Berhongaray et al., 2017). The most reliable method for measuring the carbon balance of willow stands is likely the eddy covariance method, which is not feasible in experiments with small plots. Part of the uncertainty also arises from the simplicity of the models. For example, soil respiration was modelled only based on soil temperature and WTD, although it can also be affected by, for example, changes in microbial community composition or activity (Yang et al., 2022) and soil moisture, which does not always follow changes in WTD well (Smith et al., 2018). Estimating vegetation cover using measured LAI is also problematic as it reflects weakly the amount of active chlorophyll (Delegido et al., 2015; Gregersen et al., 2013). It is especially difficult to assess active vegetation at the beginning and end of the growing season. However, the influence on the annual balance is minor due to low temperature and radiation at that time. With the Canopeo application, the models were significantly better as it was possible to determine the green leaf area better than with the previously used LAI measurement with the SunScan instrument. The measurement results of PAR values feature uncertainties due to abrupt changes in cloudiness or fogging and dirt on the plexiglass. Due to technical problems, FMI data and another PAR sensor were used to fill the gaps in the PAR measurements, especially in 2021. The plexiglass surfaces were kept as clean as possible; fogging was kept low using a short measurement time; and clear-sky conditions

were preferred, which should reduce the uncertainty occurring in measurements. Model-predicted soil temperature in gap filling may cause some error, but the filled gaps were not long, and the error was mostly diurnal with low significance for the annual balances. Regarding biweekly $N_2O$ and $CH_4$ measurements, there is a high risk of missing short-term peaks, for example, due to freeze–thaw cycles (Lammirato et al., 2021). Also, if the measurements hit peaks, the emissions may be overestimated due to the interpolation of the gaps in the data, particularly during times with infrequent measurements.

## 5  Conclusions

This study gave valuable insights into the practical implementation and climate mitigation potential of three management options relevant for cultivated peatlands with raised WTD: forage, willow and set-aside. The results indicate that wet management of cultivated peat soils considerably reduces the soil respiration and $N_2O$ emissions. Significant counteracting effect of increased $CH_4$ emissions is avoided as long as the WTD does not rise close to the soil surface. However, compared to full rewetting, partial rewetting remains a compromise solution to climate warming as it is likely that the peat layer will eventually be lost. It is important to develop incentives to inundate large, connected peatland areas to ensure water availability and maintenance of a high enough water table for efficient control of peat decomposition.

**Data availability.** The data will be available on Zenodo: https://doi.org/10.5281/zenodo.14142546 TS2.

**Supplement.** The supplement related to this article is available online at: https://doi.org/10.5194/soil-10-1-2024-supplement.

**Author contributions.** KL and HK designed the experiment. JH, HH, SS and TL developed the methodology. HK and TL planned, supervised and partly conducted the fieldwork. KL, HH and HK analysed and visualised the data. KL and HH wrote the original manuscript. All authors were involved in revising the text.

**Competing interests.** The contact author has declared that none of the authors has any competing interests.

**Special issue statement.** This article is part of the special issue "Trade-offs and synergies of soil carbon sequestration and environmental impacts: implications for agricultural management". It is a result of the EGU 2023, Vienna, Austria, 23–28 April 2023.

**Acknowledgements.** The authors are grateful to the technical staff of Natural Resources Institute Finland for their skilled work in the field and laboratory.

**Financial support.** This research has been supported by the Research Council of Finland (grant no. 312912) and Horizon 2020 (grant no. 862695).

**Review statement.** This paper was edited by Ana Meijide and reviewed by two anonymous referees.

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

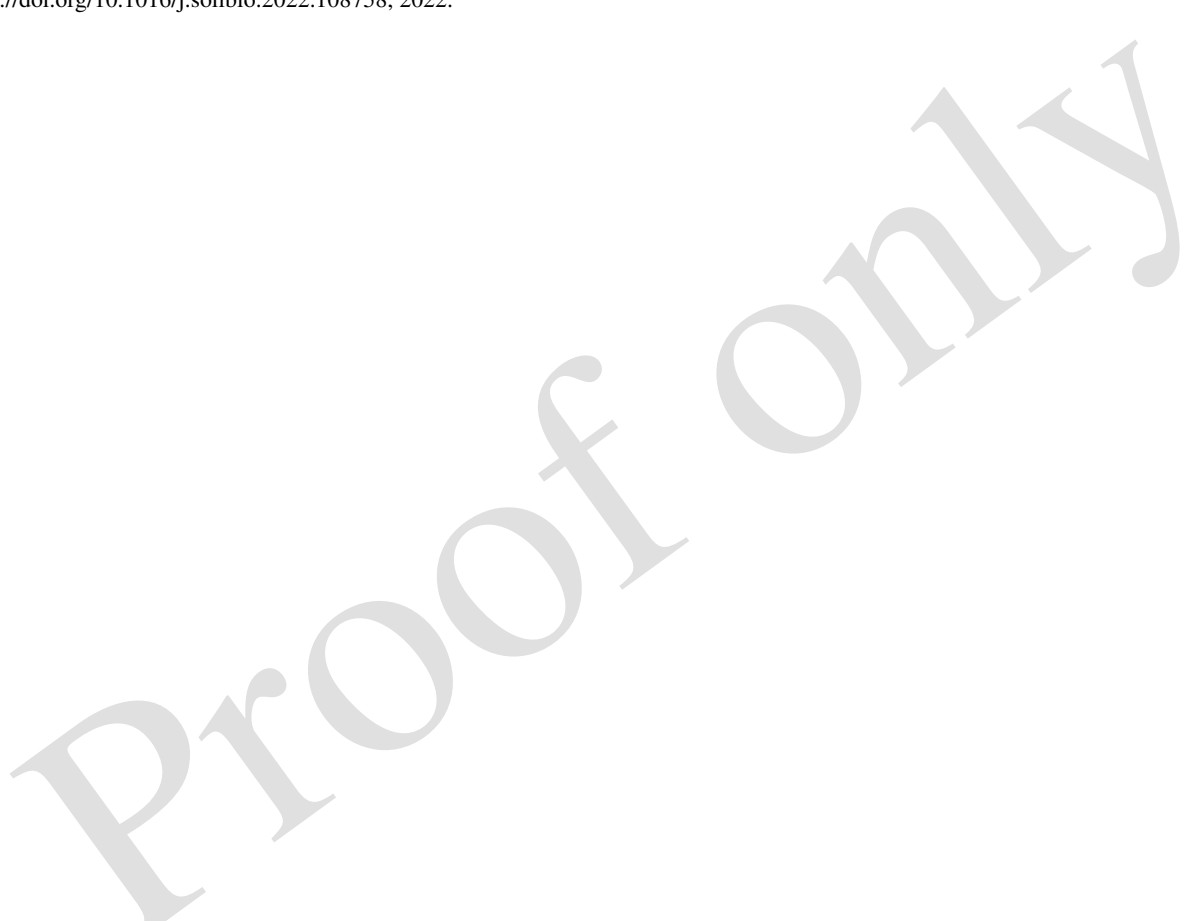

**Remarks from the typesetter**

**TS1** We have not adjusted the value here and on page 2. Meaning and content changes, including changes to values, should be reviewed by the editor before being implemented in the proofreading stage. Please reassess if these changes are strictly necessary before taking this step. For more information, please see our proofreading guidelines at: http://publications. copernicus.org/for_authors/proofreading_guidelines.html. If you want us to change the equations, please prepare an explanatory document (doc or pdf) which we can send to the editor via our system. If you insist on these changes we have to forward your requests to the handling editor for approval. To explain the corrections needed to the editor, please send me the reason why these corrections are necessary. Please note that the status of your paper will be changed to "Post-review adjustments" until the editor has made their decision. We will keep you informed via email.