# Peer review of "Impact of crop type on the GHG emissions of a rewetted cultivated"

_EGUsphere, 2024_

## Author Response (AR1)

**R1**

**General comments:**

Kristiina Lång and co-authors present data from four years of measuring greenhouse gas emissions ($CO_2$, $N_2O$, $CH_4$) of a (moderately) rewetted former agricultural field site in southern Finland. Three treatments were established (willow, forage, set-aside (bilberry was planted but didn't grow)). The presented dataset comprises greenhouse gas flux data for four years. In addition, a full C balance including photosynthesis, ecosystem respiration, net ecosystem exchange, carbon in the plant biomass, and net ecosystem carbon balance was calculated. Overall, it is a very detailed dataset which will certainly be of interest for biogeochemical modeling of greenhouse gas emissions from rewetted peatlands. The paper is well written and easy to read. My main remarks are:

- A knowledge gap / description of the relevance of the study is missing.

- Methodology is not always clear (see specific comments).

- The authors did a lot of modeling on measured fluxes, but it remains unclear why and what the additional knowledge gain is.

We thank the reviewer and respond to the questions below.

**Specific comments:**

The introduction nicely summarizes the current knowledge about greenhouse gas emissions from peat soils and the effects of rewetting and changing water table depths. The introduction is followed by three hypotheses but misses a description of the research gap. After reading the introduction, it remains unclear why this study was necessary and what knowledge the authors attempt to gain. Similarly, results are mostly discussed separately in relation to other studies. There's only little comparison between treatments and the overall significance of the (new?) findings remains unclear. The end of the abstract hints towards the general challenge, but also misses the description of a research gap/challenge.

- Reply: We added description of the research gap in the second last paragraph of the introduction. "Solutions for paludiculture implementation are e.g. forage and willow that can be produced in wet conditions because their roots improve the bearing capacity of the peat and thus ease machine work in wet conditions. Compared to restoration to natural conditions, paludiculture leads to compromises, as both ecosystem services and economic productivity are expected to be maintained, and it is not well known how these two aspects are best harmonised in practice. Set-aside is often not a planned management option, but wet fields drift to non-productive use when the drainage system degrades, and there are limited data on the GHG balance of such fields."

The description of the greenhouse gas measurements in the methods section is not easy to follow. It would help to get an overview of the different measured parameter first and then a description of the sampling techniques and different chambers applied. It could also help to include a figure summarizing how GP, ER, NEE, and NECB relate to each other. This could also help to explain why some fluxes have negative signs. Which is not consistently used throughout the manuscript: sometimes GP fluxes/values are described as positive values, sometimes as negative values (compare Fig. 1, Tab. 3, and L294ff).

- Reply: We changed the structure of the method description as suggested. The minus sign was added to the text starting line 294.

It is unclear why some parameters were not assessed for willow (see Tab. 3). This is not described in the methodology.

- Reply: The measurement method for willow was different from the other treatments because the willows were tall and did not fit the chambers. We clarified this by adding a new paragraph "2.6. GHG balance" in which we describe the differences in the estimation method.

A lot of outlier removal was done. It is well described and mostly justified. However, if you have 9 high values in one plot (L236), there might be an explanation to that and it feels strange that you simply decided to exclude these values.

- Reply: We agree that there may be a reason like roots growing to the unvegetated chamber area but 9 out of 601 is still only 1.5% of the total and it is difficult to justify different selection criteria for one chamber.

In addition to flux measurements, a lot of modeling was done. However, it's neither part of the introduction, nor of the hypotheses. Also, the importance/relevance of the modeling approaches in addition to measuring is unclear as it is not described.

Reply: We added text in the new paragraph 2.6 on GHG balance to clarify that modelling was used to fill the gaps between measurements in the case of CO2 as its fluxes vary a lot diurnally and thus cannot be linearly interpolated between the measurement occasions. The suggested text is "The annual net ecosystem carbon balance was constructed as the sum of the hourly values of NEE and yield data for each year in the case of forage and set-aside treatments. Modelling was used to fill the gaps between the measurement occasions to create a continuous series of hourly values. For willow, annual estimates of carbon loss in soil respiration were available from the chamber measurements from the unvegetated frames but as the willows were too high for the chambers their net production had to be estimated based on biomass accumulation during four years (Pacaldo et al., 2014). The presented net ecosystem carbon balance thus is the sum of average annual $CO_2$-C from soil respiration and average annual

amount of carbon bound in the biomass during the four first cultivation years. The cumulative annual fluxes of $CH_4$ and $N_2O$ for each management practice were calculated by interpolating the emissions between consecutive sampling days. Global warming potentials 27 and 273 were used for $CH_4$ and $N_2O$, respectively, to convert the results to $CO_2$ equivalents for the total GHG balance (Forster et al. 2021)."

-

The description of the vegetation in the set-aside plots is not clear. It is written that 'The number of species in the set-aside plots was determined once in the summer 2021' (L80). While in L276ff there is a species number for summer 2022. It's also unclear how there can be 19 different vascular plants in 18 different plant species.

- Reply: The number of plant species in all twelve plots was determined in summer 2021. In set-aside plots there were 19 vascular plants identified in 2021. The passage has now been revised in the methods and results sections.

**Technical corrections:**

Graphical abstract: shouldn't that be < 30 in the figure?

- Yes, corrected.

L13     11kg $CH_4$ à what area?

- Per hectare, corrected.

L43     not clear that paludiculture is always on peat soils

- "on peat soils" was added

L46     regarding

- Corrected.

L51     what is an oxidized layer? Please add an example

- This was revised to "oxidised, non-waterlogged, layer on the peat surface to facilitate microbial oxidation of CH4"

L114    reference for Canopeo app

- We added the reference to a scientific paper (Patrignani and Ochsner 2015). The same research organization produced the app.

L127    please add information why these different chambers were used

- All paragraphs with chamber description were amended with some more information of the parameter measured.

L148   what is the Vaisala GMP-343 probe measuring?

- "for CO2 measurement" was added.

L151   why is this done? (the shading)

It is done to gain measurement results in as many light conditions as possible to allow for reliable hourly modelling (larger range of light conditions --> more data points in modelling) to fill the gaps between measurements. The text was revised as "Net ecosystem exchange (NEE) including photosynthesis and respiration of the soil and plants was measured approximately every two weeks during the growing season using a transparent chamber ($60 \times 60 \times 60$ cm) made of polycarbonate plexiglass (1 mm, light transmission 95%). The chamber was equipped with a Vaisala GMP-343 probe for $CO_2$ measurement and a temperature and humidity sensor (Vaisala Oy, Vantaa, Finland) and two fans for mixing the air during the measurement. PAR was measured with LI-190 quantum PAR sensor (LI-COR, Lincoln, Nebraska, USA) inside the chamber. Four measurements with different amount of entering light were taken from each plot on each measurement day in order to cover a large range of light conditions and facilitate the gap filling by modelling. One or two layers of a white fabric shroud and one blackout curtain were used to acquire measurement results in different light conditions (approximately 100%, 50%, 25%, 0% of ambient radiation). The measurement with 0% radiation gave the estimate for ecosystem respiration (ER). The measurements were done in the same collars as the opaque chamber measurements. Each measurement took one minute with a five second sampling rate, or two minutes in early or late growing season when the change in flux was minor. The chamber was flushed after each measurement to reconstitute ambient $CO_2$ and air humidity contents. After closing the chamber, a lag time of 10 seconds was applied to exclude the time when the flux was not yet stabilised. Clear sky conditions were preferred to avoid problems related to changing cloud cover and to achieve the widest possible range of available light. The temperature change inside the chamber was less than 1.5 degrees which was also used as a criterion for data filtering."

-

L178   equation 2 does not match description

The description was improved; the suggested text is "Gross photosynthesis (GP) can be determined as the difference of NEE and ER. Instantaneous GP was estimated for each measurement occasion by (equation 2)"

L186   what about willow?

- The approach for willow is now explained in the new paragraph "2.6 GHG balance".

L235    leading to

- Corrected.

L318    why are results not shown? Could be in the supplementary

- These hourly values were added to the text to illustrate the range of diurnal variation but they are not maybe necessary for the supplement as the daily variation is more interesting in a study with a long measurement period. The measurement data points shown for ER in Fig. 1 already give a picture of the range of variation in the measurements.

L369    why are results not shown? Could be in the supplementary

- The results are taken from tables 2 and 3, so actually they are shown. The description was improved: "For willow, the annual NECB cannot be calculated but based on the four-year estimate on carbon binding in the biomass and carbon exported in the harvest divided to a single year, together with the average annual soil respiration and N2O and CH4 fluxes, the average annual climate impact of willow cultivation was 10.2 Mg ha-1 yr-1. "

L371ff  headline numberings not consistent, headings in discussion are meaningless

- Headings were removed.

L375    what crop was grown in the nearby field?

- Oats, it is now added to the text.

L421    comparable number

- Corrected.

Tabe 2: please explain STD

- Done

Figure 1: panel letters missing, description of panels doesn't match y-axes, please check if colors are color-blind-safe?

- The figure was revised and to our understanding red and blue are color-blind safe.

Figure 2: panel letters are huge in comparison to the font in the figure. Where do the fitted values come from? What's their level of significance? Not clear from the description.

- The panel letters were shrunk. The legend was revised to "Plot-wise mean annual fluxes…" to indicate that these are the actual measurement results from each plot.

Figure 3: unclear is it important to show the variation in $CH_4$ fluxes in the first two years (first panel a) but not in the remaining time (second panel a) or $N_2O$?  Broken y-axes might be a solution

- We tried broken y-axes for CH4 but a decent solution was not found because the differences in the flux rates are huge and it is impossible to break the axis so that all values (also the intermediate) remain visible. We hope that this solution is suitable for the journal.

R2

The authors have measured GHG fluxes and carbon balance in a Finnish peatland site that has previously been cultivated as agricultural land as has recently been somewhat rewetted by reducing drainage. The study design includes three different vegetation options and utilises several measurement techniques to determine fluxes in and out of the soil. Overall, the manuscript is well written and the study design interesting. However, I do think the manuscript would benefit from some improvements, mainly to the justification of the research topic and methodology.

- We thank the reviewer and reply the comments below.

**General comments:**

Abstract:

This is somewhat a question of preference but I would encourage writing at least something of the justification of this manuscript to the abstract as well. Half a sentence at the end of the abstract is quite little.

- We added "Raising the water table is an effective way to abate greenhouse gas emissions from cultivated peat soils" in the beginning of the abstract.

Introduction:

The general background of paludiculture in cultivated peatlands is well explained but the specific justification of this paper is somewhat missing. I would have wanted to know, here or in the methods, why you chose forage, willow and set-aside (although this was a little bit better explained) options.

- We added description of the research gap in the second last paragraph of the introduction. "Solutions for paludiculture implementation are e.g. forage and willow that can be produced in wet conditions because their roots improve the bearing capacity of the peat and thus ease machine work in wet conditions. Compared to restoration to natural conditions, paludiculture leads to compromises, as both ecosystem services and economic productivity are expected to be maintained, and it is not well known how these two aspects are best harmonised in practice. Set-aside is often not a planned management option, but wet fields drift to non-productive use when the drainage system degrades, and there are limited data on the GHG balance of such fields."

L41: Furthermore, you state that $CH_4$ emissions do not compromise the mitigation potential according to literature which makes it sound like there isn't much to research here, yet it's still a research question in this manuscript. I would expand a little bit on this topic, seen as it's one of the research questions (and certainly an interesting one at that).

- We continued the sentence in the introduction with "...but data is needed to understand the factors regulating CH4 emissions that can sometimes be high after rewetting (Nielsen et

al., 2023)" as sometimes methane emissions are high after rewetting and measurements in different environments are still needed to understand the phenomenon.

Materials and methods:

I had a little trouble keeping in mind which areas the terms "plot, site, block and experimental area" refer to. Firstly, in L104, I'd refer specifically to the site instead of experimental area if WTD was indeed only measured from the corners of the whole site, to keep it simpler. Secondly, I know the graphical abstract should be as simple as possible, but a graph showing the study design in more detail and explaining "plot", "block" and "site" would really help.

- "Plot" was added to the graphical abstract. Experimental area was changed to site in line 104 (the original pdf) and now we only use the term "site" for the whole area. The description in the method section was changed to better describe the site: The site was established in 2018 and it consists of twelve experimental plots (9 × 6 m) in four blocks (see the graphical abstract).

L69: Considering how important rewetting is for mitigating the effects of cultivated peatlands, I would have liked to see a little more information on the rewetting method itself. Now, the only information is that a control well was installed but could you maybe explain this in a bit more detail.

- We added "The adjustable tube inside the well was set to a position letting the water out when the water table reached 20 cm depth below the soil surface."

L104-114: There were some changes made to the ancillary measurement techniques during the study campaign. This is understandable, but did you estimate if these, namely changing the WTD measurement location and swapping LAI to green canopy cover, have any effect on the results? It would also be nice to see, for example in an appendix, how the green canopy cover responds better to photosynthesis. Now this sentence was rather brushed aside.

- Unfortunately we did not compare the WTD results made at each plot and in the corners of the site.
- We also did not make measurements of LAI and green canopy cover at the same time and thus have no data for the comparison.
- We added reference to a paper comparing Canopeo results to 2 other methods to measure canopy cover (Govindasamy et al. 2022).

I'm a little confused about the modelling work here. Could you at least explain why the flux modelling was done? I would also expect to see some estimation of how the modelled fluxes compared to measurements in addition to stating the modelled and measured maximum values.

- We added text to explain that modelling was needed to fill the gaps between measurement days. A sentence on that was added in 2.4 Flux modelling "The gaps in GP and ER data between the measurement occasions were predicted using hourly timeseries of the ancillary data."
- Also, the descriptions of modelling were generally modified to make the process clearer.

Results:

Fig 1: Please increase the text size. For example, moving the y-axis text to each plot's headline and keeping only the unit on the y-axis would already make this much easier to read. Furthermore, is there a good reason why all the plots don't show all treatments? For temperature and PAR this makes sense, but why is there no black line for vegetation index and daily mean GP? I'm also not sure why you haven't included the measurements for daily mean GP. For WTD, are these results taken from the four corners of the site for the whole period or are the plot measurements included in 21-22? If the latter, I'm not sure this is a good way to show the results as it seems like there is more variability in the later years. If the former, is there another reason for the increased variance?

- The text size was increased. The black line represents the willow for which the method of estimating the carbon balance differed from the others and thus it's components cannot be shown in this figure like for the other treatments. This is now explained in the new paragraph 2.6. GHG balance. Daily GP was not measured but estimated based on the measured values of NEE and ER and we have tried to improve the description of this. All WTD measurements are included in the site mean and the greater variation in the latter years results from more frequent measurements due to continuous measurement using loggers instead of manual measurement every other week.

Table 3: Is there a reason why table 3 shows only some of the results for willow? This was perhaps addressed somewhere but I couldn't find it, and perhaps the reason could be reiterated in the table caption.

- This is now explained in the paragraph on GHG balance and we suggest adding a footnote below the table "[a]All components of the carbon balance are not available for willow, see chapter 2.6"

L350: I would really prefer to see similar figures to Fig 2 from both crop vs. $CH_4$ and year vs. $CH_4$ although this can easily be put to the appendix (and combined as well). With such a low n-value, I don't think it's sufficient to say that these had no effect on $CH_4$ fluxes only based on the p-value, particularly when in table 3, there does seem to be a clear relationship between $CH_4$ and year. There is such a strong change during the years from a sink to a source that a very simple statistics might not

capture the relationship but it doesn't mean it's not worth showing. I would also like to see some other metrics for measuring the relationship between these variables besides p-value. The same applies to the $N_2O$ fluxes where there are relationships between the variables but only the p-value is given. Fig 3 does provide some of this information but it is very difficult to read and to distinguish differences particularly between the vegetation options.

- Boxplots were created for the supplement (Figs. S3-4) to show the variation between years and crops. The change from methane sink to source is now well visible from the figure, as well and the decrease in N2O emissions. We actually did not expect differences between crops in this study as the WTD is clearly the most important variable regulating the emissions and that's also shown in Fig. S4. We added some more results on the mixed model analyses in the supplement too (Table S4).

Discussion:

I would somewhere like to see some discussion related to the rewetting. You state in the introduction that the target WTD was reached only periodically but could you explain why that is?

We suggest adding the following discussion on this. "The target WTD was not reached for most of the time likely because there was unexcepted lateral water outflow from the site. Our strategy of raising the WTD at a limited area within a field parcel was thus not successful. The larger the area where the water outflow is restricted, the better the result likely is, and catchment level water management planning is often recommended for the best results (Mitsch and Wilson, 1996; Pasquet et al., 2015). "

Conclusions:

The last sentence of the conclusions is a bit surprising as I'm not sure that this (while certainly true) is strongly related to or clearly visible from this work.

- We tried to make this clearer with the discussion on the success of the rewetting (see the previous comment).

**Specific comments:**

Graphical abstract: Could you explain why the graphical abstract doesn't show any $CH_4$ emissions? According to figure 2 and section 3.3, there should be a relationship between WTD and $CH_4$.

- The methane fluxes are not visible in this scale.

L43: Adding "i.e." before "crop production" might make the sentence a bit easier to read.

- Done.

L46: "As regards **to** GHG mitigation"

- Done

L101-102: Could you add references to both the FMI station and the global radiation data product?

- The current reference is formulated as required by the data producer.

L108: Could you explain what "WTD was measured manually from monitoring pipes when possible" means?

- We changed the text to "…when the water was not frozen."

L117: no green vegetation **was present**

- Done.

L233: Fig 2 should probably be Fig 1.

- Yes, it should be Fig. S1. Corrected.

L268: As snow is not an important variable in this manuscript, this is a bit of a nitpicky comment, but are the snow days calculated from a calendar year or from each snow season? To my mind, calculating calendar year snow days doesn't make much sense as it gives very little information on the impact of snow cover on any ecosystem processes.

- The sentence was revised as "Number of days with a snow-cover on the soil within each modelling year (April to March) was 13, 81, 108 and 118, respectively."

L379: probably should be "rised" and not "raised"

- Corrected.

L467: "considerab**ly**"

- Corrected.

L468: "WTD does not r**is**e"

- Corrected.

---

## Author Response (AR2)

Replies to the comments for the second revision

We thank the reviewers for reasonable and useful comments and try to reply below to the latest requests.

Methods:

Concerning the description of the vegetation index, I have two comments:

1. In the response letter, you say that you have added a reference by Govindasamy et al. to section 2.2. I was not able to find this reference, so please make sure to include it.

    *- We are sorry, this was forgotten in the last revision. Now we found a different reference that includes measurements with better comparable methods (Shepherd et al. 2018). (Line 127+list of references)*

2. I also asked you to further clarify, how the green canopy responds better to photosynthesis. I do not think "it likely responds better to photosynthesis" is a sufficient clarification. I am sure you are correct here but please provide some concrete justification for this.

*- The text was revised as: In 2022, we measured green canopy cover with the Canopeo app (Patrignani and Ochsner, 2015) instead of LAI. Based on our experiences, and due to the operation and physical design of the LAI device, it did not provide as comprehensive picture of the biomass inside the gas measurement collar as Canopeo. Vegetation index has been found to be faster to measure and less dependent on the ambient light conditions than the light interception method (Shepherd et al. 2018). (Line 123)*

I think the flux modelling is now better explained. However, I had mentioned that I would typically expect to see some kind of estimation between modelled and measured fluxes to understand the uncertainty. You hadn't replied anything to this comment, so I raise it again here. If you don't think it's necessary to provide this, I'd at least appreciate an explanation in a response letter.

- *Figure S1 provides the comparisons between modelled and estimated GP and ER. There are also the measurement points of ER added to Fig. 1, so that the reader can compare the modelled and measured values. We are sorry for not stating this out in our previous reply.*

Concerning references to the FMI weather station and global radiation data set: I don't think a Creative Commons license means that a citation is not needed, it should only mean that the data can be freely used (with appropriate credits to the author/data provider). Additionally, I do think it's also useful to provide the information of which FMI weather station was used.

- *The name of the weather station was added (Line 111) and the reference to FMI open data as well (Line 111+list of references).*

Results:

Concerning this comment from last round:

"L350: I would really prefer to see similar figures to Fig 2 from both crop vs. CH4 and year vs. CH4 although this can easily be put to the appendix (and combined as well). With such a low n-value, I don't think it's sufficient to say that these had no effect on CH4 fluxes only based on the p-value, particularly when in table 3, there does seem to be a clear relationship between CH4 and year. There is such a strong change during the years from a sink to a source that a very simple statistics might notcapture the relationship but it doesn't mean it's not worth showing. I would also like to see some other metrics for measuring the relationship between these variables besides p-value. The same applies to the N2O fluxes where there are relationships between the variables but only the p-value is given. Fig 3 does provide some of this information but it is very difficult to read and to distinguish differences particularly between the vegetation options."

I appreciate the boxplot in supplements. However, I don't think the table S4 is very useful as it is in providing information about the linear effect models. It is very difficult to interpret as there is no additional information given about the variables in the table and it isn't linked in any way to the primary text. As I stated previously, I'm not convinced of the usefulness of only providing p value, so I think a little more analysis of the results of the linear mixed models would help the readers.

- *It was not very clear if the reviewer recommended deleting Table S4 or adding information to it. We added the information if the dependent variable was log-transformed or not in Table S4 to facilitate further use of the results e.g. for building equations based on the effect values. We also improved the explanation in the methods section on building of the models (Line 286). There would not be much relevant information added if the whole result set of each model was added in the supplement as the technical issues are explained in the methods, thus we hope that this solution is satisfactory.*
- *We have now linked table S4 better to the text by adding references to the table in the text (Starting line 326), with some additional aspects that can be said based on the mixed model analyses. It is true that the models or variables are not explained in the table but the model building process and method to acquire the values is described in the methods and we added in the description of statistical methods also the principle that all relevant variables were first included and then one by one removed if they were not significant.*
- *The largest change was done to the description of the mixed model for GP, ER and NECB as in their case we noticed that the effect of WTD was actually misleading in the*

*models that were originally selected. The previous model suggested that raise in WTD increases these variables, but the increasing trend likely resulted only from an increase of biomass as the experiment aged. It is now explained in the text that a mixed model without WTD is better (also based on the Akaike's information criteria). The effect of WTD on the most relevant variables, the gas fluxes, is illustrated in figures 2 and S2 separately.*

- *In the case of soil respiration, figures 2 and S2 provide the best information on the effect of WTD but we added more detail on the effect of crop type on respiration (Line 371).*
- *In the case of methane and nitrous oxide, we added references to Fig S3-S4 and Table S4 which should make it easier to understand the results.*

Finally, a couple things about your response in general: Reviewing the modifications that you made would have been considerably easier, had you included the line numbers of where each modification was done in your author response. Furthermore, please make sure that the tracked-changes -version corresponds with the original manuscript. Now, lines 196-200 appeared out of nowhere to the new version which didn't exactly help.

- *We are sorry for the sloppy work! We tried to be more careful in this round.*

*Some typos were corrected and some additional improvements made:*

- *Line 236: Two sentences on soil respiration estimation were added.*
- *Line 265: Reference to Matlab was added.*
- *Table S3 was revised as there were results of 2 alternative model parametrizations. Now only the final version remains.*

---

## Author Response (AR3)

Dear Editor,

We modified the below version from the suggested title versions.
Impact of crop type on the GHG emissions of a rewetted cultivated peatland

No other changes were made to the manuscript. Thank you for your cooperation.

Kind regards,

Kristiina Lång